# Molecular Targets Implicated in the Antiparasitic and Anti-Inflammatory Activity of the Phytochemical Curcumin in Trichomoniasis

**DOI:** 10.3390/molecules25225321

**Published:** 2020-11-14

**Authors:** Natalia Mallo, Jesús Lamas, Rosa Ana Sueiro, José Manuel Leiro

**Affiliations:** 1Department of Microbiology and Parasitology, Laboratory of Parasitology, Institute of Research on Chemical and Biological Analysis, Campus Vida, University of Santiago de Compostela, 15782 Santiago de Compostela, Spain; mallo.natalia@gmail.com (N.M.); rosaana.sueiro@usc.es (R.A.S.); 2Department of Fundamental Biology, Institute of Aquaculture, Campus Vida, University of Santiago de Compostela, 15782 Santiago de Compostela, Spain; jesus.lamas@usc.es

**Keywords:** *Trichomonas vaginalis*, curcumin, hydrogenosomal enzymes, proteinases, proinflammatory cytokines

## Abstract

Trichomoniasis, is the most prevalent non-viral sexually transmitted disease worldwide. Although metronidazole (MDZ) is the recommended treatment, several strains of the parasite are resistant to MDZ, and new treatments are required. Curcumin (CUR) is a polyphenol with anti-inflammatory, antioxidant and antiparasitic properties. In this study, we evaluated the effects of CUR on two biochemical targets: on proteolytic activity and hydrogenosomal metabolism in *Trichomonas vaginalis*. We also investigated the role of CUR on pro-inflammatory responses induced in RAW 264.7 phagocytic cells by parasite proteinases on pro-inflammatory mediators such as the nitric oxide (NO), tumor necrosis factor α (TNFα), interleukin-1beta (IL-1β), chaperone heat shock protein 70 (Hsp70) and glucocorticoid receptor (mGR). CUR inhibited the growth of *T. vaginalis* trophozoites, with an IC_50_ value between 117 ± 7 μM and 173 ± 15 μM, depending on the culture phase. CUR increased pyruvate:ferredoxin oxidoreductase (PfoD), hydrogenosomal enzyme expression and inhibited the proteolytic activity of parasite proteinases. CUR also inhibited NO production and decreased the expression of pro-inflammatory mediators in macrophages. The findings demonstrate the potential usefulness of CUR as an antiparasitic and anti-inflammatory treatment for trichomoniasis. It could be used to control the disease and mitigate the associated immunopathogenic effects.

## 1. Introduction

*Trichomonas vaginalis* is a flagellated protist that parasitizes the human urogenital tract and causes trichomoniasis. Trichomoniasis is the most prevalent non-viral sexually transmitted disease worldwide and is associated with serious public health problems such as transmission of HIV [1,2]. To infect the urogenital tract, the parasite must develop adaptation systems to support the unfavorable changes that take place in this adverse environment (involving pH, temperature, hormonal changes, menstruation, oscillation in iron concentrations, etc.) [1,3]. Virulence factors such as proteinases contribute to the cytotoxicity of *T. vaginalis*, the immunopathogenic activity and evasion of the immune response during human infection [2,4,5]. Cysteine proteinases are some of the most abundant proteinase families in *T. vaginalis* and in parasites in general [2,6]. Several proteinases are reported to be involved in virulence and some are regulated by iron. These proteins have been well studied in *T. vaginalis* and are known to be implicated in most of the functions necessary for parasitism such as cytoadherence, cytotoxicity, hemolysis and immune evasion [1,3,7,8,9,10,11]. Vaginosis, in which *T. vaginalis* is implicated, it is usually associated with bacterial infections that alter both the normal cervico-vaginal innate immunity and the inflammatory response, producing increased levels of TNFα, IL-1β, IL-8 and vaginal neutrophils [12].

The protein pyruvate:ferredoxin-oxidoreductase (PfoD) is also regulated by iron in *T. vaginalis*. This is a key protein in the hydrogenosomal metabolism of the flagellate and is responsible for decarboxylation of pyruvate to acetyl-Co-A, which is thought to be involved in metronidazole resistance [13,14,15,16]. Metronidazole is the current treatment for trichomoniasis, and although it is usually effective, the number of metronidazole resistant strains is increasing [2], and new treatments are needed. Natural compounds are potential sources of alternative treatments and phytochemicals have a huge potential in drug discovery and therapy. Several studies have investigated the use of natural compounds to treat trichomoniasis [17,18]. Among such compounds, the polyphenols resveratrol (RESV) and curcumin (CUR) have been reported to display activity against *T. vaginalis* [19,20]. The rhizomes of the perennial plant *Curcuma longa* (Zingiberaceae) include in their composition curcuminoids (polyphenolic pigments) such as CUR, demethoxycurcumin and bis-methoxycurcumin, as well as volatile oils, sugars, proteins and resins; however, CUR (diferuloylmethane) is generally the main polyphenol found in turmeric and is generally considered the most active component, to which the main pharmacological activities are attributed [21]. It can act as an antioxidant, reacting directly with ROS and RNS and inducing expression of antioxidant proteins, and it can also act as an anti-inflammatory and antiparasitic agent [22,23,24]. In addition to its scavenging properties, CUR can chelate positively charged metals, such as iron, found at the active sites of several proteins [25]. This is an important factor in the growth and metabolism of *T. vaginalis* [26,27].

CUR also has anti-inflammatory properties and can inhibit glucocorticoid receptor (mGR) transcription [28], which mediates the transcription of target genes by affecting its phosphorylation status [29]. Likewise, CUR has been reported to regulate the signaling of another transcription factor, NF-κB, which is involved in cytokine expression [30,31]. CUR thus modulates the host immune response and affects parasite survival.

CUR is approved for use by the Food and Drug Administration and the Joint FAO/WHO Expert Committee on Food Additives, and relatively high dose tolerance without side effects has been reported [32].

In the present study, we searched for molecular targets on which CUR can act and that explain the antiparasitic effect of the compound. We also investigated the influence of CUR on molecules that are key to parasite survival (enzymes implicated in hydrogenosomal metabolism) and host invasion (proteinases). In addition, we investigated the modulatory role of CUR on the pro-inflammatory response induced by *T. vaginalis* proteinases in a mice macrophage model.

## 2. Results

### 2.1. In Vitro Antiparasitic Effect of Curcumin

At a concentration of 100 μM, CUR caused a significant decrease in the in vitro growth of *T. vaginalis* from the first day of culture (Figure 1), relative to control without CUR. It thus displayed cytostatic activity, with an IC_50_ of about 117 ± 7 μM on the first day; however, the inhibitory effect decreased on the second day of culture, with an IC_50_ of 173 ± 15 µM. DMSO did not affect parasite growth.

### 2.2. Influence of Curcumin on Pyruvate-Ferredoxine Oxidoreductase (PfoD) Gene Expression and the Hydrogenosomal Membrane Potential (ΔΨm)

At a concentration of 100 µM, CUR produced significant upregulation of the mRNA expression levels of the PfoD hydrogenosomal enzyme; however, lower doses of the polyphenol did not have any effect on the activity (Figure 2A).

The fluorescent probe JC-1 was used to analyze the effect of CUR (100 μM) on ΔΨm levels. The hydrogenosomal membrane potential levels decreased significantly after incubation for 2 h with the polyphenol in MDM (Figure 2B).

### 2.3. Impact of Curcumin on Parasite Proteolytic Activity

The proteolytic activity of *T. vaginalis* was altered by addition of CUR, as revealed by qualitative analysis of the proteolytic banding pattern in the SDS-PAGE-gelatin assay (Figure 3A). The pattern of the bands with proteolytic activity was very similar in treated and untreated samples; however, dark bands, which represent proteolytic activity, were wider in untreated samples, as readily visualized on the bands situated at about 50 kDa. 

Quantitative analysis by use of a gelatin-FITC fluorescence proteolytic assay revealed a significant decrease in proteolytic activity levels in the CUR treated extracts (Figure 3B), consistent with the previous results.

### 2.4. Influence of Curcumin on the Pro-Inflammatory Processes Induced by Lipopolysaccharide (LPS) and T. vaginalis Proteinases in Immune Cells

We initially investigated the effect of CUR on production of NO (a proinflammatory mediator derivative of oxidative stress) in mouse RAW 264.7 macrophages stimulated with *T. vaginalis* proteinases or with bacterial LPS (generated during bacterial vaginitis). CUR produced significant inhibition of NO production in both types of stimulated cells without affecting cell viability and cytotoxicity (see Appendix A). Inhibition was even higher than that generated by the NO synthase inhibitor LNMA (Figure 4).

We also investigated the effect of the CUR on gene expression of cellular mediators of inflammation such as the proinflammatory cytokines TNFα and IL-1β in RAW 264.7 mouse macrophages stimulated with bacterial LPS or with the parasite proteinases (Figure 5). The results indicate that the CUR (100 μM) caused significant inhibition of the proinflammatory cytokine TNFα in cells stimulated with LPS or with parasite proteinases (Figure 5A). At the same concentration, CUR also strongly inhibited IL-1β expression in LPS-stimulated macrophages; however, the *T. vaginalis* proteases generated much lower levels of mRNA than those generated by LPS, although in this case the inhibition of IL-1β expression by CUR was also significant (Figure 5B).

### 2.5. Effects of Curcumin on Other Mediators of Inflammation in Macrophages Stimulated with Lipopolysaccharide (LPS) and Proteinases of T. vaginalis

Finally, we analyzed the effect of CUR on mRNA and protein expression of the genes encoding the glucocorticoid receptor mGR and the heat shock protein Hsp70. In both cases, RAW 264.7 macrophages were stimulated with LPS or *T. vaginalis* proteinases (see Figure 6). At a concentration of 100 μM, CUR caused significant inhibition of mGR gene expression, mRNA and protein levels in stimulated cells (Figure 6A,C); however, the polyphenol had the opposite effect on Hsp70 gene expression, inducing increases in mRNA and protein levels (Figure 6B,D).

## 3. Discussion

Finding alternative treatments for trichomoniasis is important as several strains of *T. vaginalis* are known to be resistant to MDZ, the most common drug used to treat the disease. The present findings show that CUR exerts an in vitro antiparasitic effect on *T. vaginalis*, as recently reported for other strains of *T. vaginalis* [20]; however, the mechanisms underlying the effects of the polyphenol on molecular activity, which would help explain the antiparasitic activity, are not known.

Iron is an essential metal required by many organisms as a cofactor in some biochemical activities that may be essential for parasite survival in the host, such as virulence factors [3]. The hydrogenosome of *Trichomonas* is characterized by the presence of iron-sulphur (FeS) cluster-containing enzymes such as pyruvate: ferredoxin oxidoreductase (PfoD), which is regulated by iron [26,27,33]. We observed upregulation of mRNA expression levels in *T. vaginalis* trophozoites treated with CUR, and it is therefore possible that the antiparasitic effect of CUR on *T. vaginalis* involves Fe transport, as CUR has been described as a metal chelator [25]. CUR may also affect the fermentative energetic metabolism of *T. vaginalis* as a result of its antioxidant properties, because it acts via the induction of hydrogenosomal dysfunction, as revealed by use of a JC-1 probe to study the hydrogenosomal membrane potential levels, which were downregulated by CUR and which may be a signal of cellular death.

Some of the parasite proteins related to cellular death and apoptosis are proteinases [2,34]. Experimental animals inoculated with whole *T. vaginalis* preparations produced antibody to many proteinases, indicating the immunogenic nature of trichomonad proteinases and their importance in parasite virulence and pathogenesis [35]. As with other parasites, *T. vaginalis* proteinases have been described as virulence factors, and the chemotherapeutic potential of proteinase inhibitors has been highlighted [9,34,36,37]. CUR has been reported to prevent increased proteolytic activity in mice and rat muscle [38]. In this case the anti-inflammatory properties of CUR [39] act on parasite proteolytic activity, particularly on the production of NO and as modulator of the gene expression of the proinflammatory cytokines, TNFα and IL-1β [40], induced by parasite proteinases.

Nitric oxide (NO) production by the NO-synthase isoform NOS-II has been specifically described to play an important role as an effector of macrophage-mediated cytotoxicity and immune modulation against numerous parasites, including *T. vaginalis* [41,42]. Some investigators attribute the NO cytotoxicity to the iron-scavenging properties of the nitrogen by-products, resulting in iron depletion and the consequent inactivation of iron-requiring molecules [41]. However, other investigators attribute NO cytotoxicity to the inhibition of cysteine proteinases of several parasites, such as *Plasmodium falciparum, Trypanosoma cruzi* and *Leishmania infantum*, by NO-releasing compounds [42]. The antioxidant properties of CUR have been defined as the ability of the compound to scavenge NO [43]. We observed this effect in mouse peritoneal macrophages in which NO production was activated by the addition of LPS and *T. vaginalis* proteinases, and it was even more pronounced than that caused by the NO-synthase inhibitor LNMA. 

Interleukin 1β (IL-1β) and tumor necrosis factor α (TNFα) are important proinflammatory cytokines that can trigger inflammatory signaling pathways by activating nuclear factor κB (NF-κB) (a transcription factor that can be stimulated by stress). CUR, like other plant-derived phytochemicals, has the ability to target the NF-κB signaling pathway [23,30,44,45,46]. We observed that addition of CUR significantly downregulated expression of TNFα and IL-1β genes in macrophages stimulated with LPS or *T. vaginalis* proteinases. 

The glucocorticoid receptor (mGR) is also associated with the chaperone complex. This ligand-dependent transcription factor mediates developmental and metabolic processes in response to glucocorticoids and acts as a regulator of the promoters of glucocorticoid responsive genes as well as other transcription factors involved in inflammatory responses and cellular proliferation. mGR is bound to the chaperone complex in its cytoplasmic state, before ligand binding, and upon ligand binding it translocates to the nucleus where it binds specific DNA sequences and modulates transcription [29]. mGR-mediated transcription has been found to be inhibited by CUR via phosphorylation [28,29]. In the present study, we found that the addition of CUR to mice macrophage cells inhibited upregulation of mGR protein and gene expression levels by LPS and *T. vaginalis* proteinases, again indicating the anti-inflammatory properties of this polyphenol.

TNFα and IL-1β are negatively regulated by heat shock factor (HSF-1), a transcription factor that drives the increase in heat shock proteins (Hsps) in response to stress. The relationship between NFκB, Hsps and HSF-1 is complicated, and activation of NFκB increases significantly as HSF-1 decreases. Consequently, independently of the activation of Hsps expression, HSF-1 inhibits transcription of a number of pro-inflammatory genes and modulates the activation of NFκB [45,47]. Heat shock response supports cells by regulating physiological homeostasis under stress conditions, such as in a pathological disorder. CUR, which downregulates the pro-inflammatory and proliferative capacities of *T. vaginalis,* induces a heat shock response, as reported in several organisms, including human cells [46,48,49,50]. Some natural compounds are known to be capable of activating chaperone induction and thus interfering in cell survival [46,50]. In the present study, we observed an increase in Hsp70 protein and gene expression levels in response to addition of CUR along with stress stimuli such as *T. vaginalis* proteinases and LPS.

In conclusion, CUR exerts an in vitro anti-parasitic effect on *T. vaginalis* by affecting the hydrogenosomal function. CUR also exerts an anti-inflammatory effect on the host, via the pro-inflammatory effect of the parasite proteinases, demonstrating the potential suitability of this compound as a treatment for trichomoniasis and for improving the immunopathological effects derived from the disease. 

## 4. Materials and Methods 

### 4.1. Parasites 

*Trichomonas vaginalis* isolate Tv1 was obtained from a patient with vaginal trichomoniasis attending the gynecology service of the Santiago de Compostela University Hospital Complex (Spain). Isolates were cultured axenically in vitro, in modified Diamond’s medium (MDM) containing 2% (*w*/*v*) trypticase, 1% (*w*/*v*) yeast extract, 0.5% (*w*/*v*) maltose, 0.1% (*w*/*v*) l-ascorbic acid, 0.1% (*w*/*v*) l-cysteine, 0.1% (*w*/*v*) KCl, 0.1% (*w*/*v*) KHCO_3_, 0.1% (*w*/*v*) KH_2_PO_4_, 0.1% (*w*/*v*) K_2_PO_4_ and 0.02% (*w*/*v*) FeSO_4_ (pH 6.2) supplemented with 10% (*v*/*v*) heat-inactivated fetal bovine serum, as previously described [19,51]. Trophozoites were cultured at 35 °C in 15-mL sterile tubes completely filled with medium to create an O_2_ poor environment. Cells were grown to late log phase (1 × 10^6^ to 2 × 10^6^ cells/mL) and harvested by centrifugation (5 min at 200× *g*) for use in all experiments [19].

### 4.2. Culture of Murine Macrophages 

Macrophage-like cell line RAW 264.7, derived from BALB/c mice, was acquired from the American Type Culture Collection (ATCC; cat. N TIB-71). The cells were cultured following the supplier’s instructions, in Dulbecco’s modified Eagle’s medium (Sigma-Aldrich, Madrid, Spain) supplemented with 10% inactivated fetal bovine serum (FBS) and 1% of an antibiotic/antimycotic solution (100×) (100 units/mL penicillin G, 0.1 mg/mL streptomycin) at 37 °C in humidified atmosphere containing 5% CO_2_ [52].

### 4.3. Antiparasitic Activity 

The effect of CUR (Sigma-Aldrich) on the in vitro growth of *T. vaginalis* was analyzed as previously described (19). Briefly, different concentrations of CUR (50, 100 and 125 μM) were added to the cell cultures from a 100 mM stock solution prepared in dimethyl sulfoxide (DMSO) and stored in the dark at −20 °C until use. Parasites were cultured in sterile 24-well culture plates (Corning, New York, NY, USA) containing 2 × 10^5^ trophozoites/well in MDM with 5 replicates of the different treatments for 2 days at 35 °C in a container under a vacuum. A control containing the highest concentration of DMSO used was included in the assay. The concentration of trophozoites was determined daily by counting the cells contained in 10 μL aliquots of the cell suspensions removed from each well. Cells were counted in a hemocytometer. The in vitro half maximal inhibitory concentrations (IC_50_) (in relation to the antiparasitic activity of CUR) were calculated by regression analysis. An Excel program (Microsoft Office software package) was used to implement the linear regression formula y = mx + b, where m is the slope y_1_ − y_2_/x_1_ − x_2_ and b the intercept value of the line by using the equation: IC_50_ = (0.5 − b) × log dose/m.

### 4.4. Hydrogenosomal Membrane Potential Assay 

The hydrogenosomal *T. vaginalis* membrane potential (ΔΨm) was determined using the JC-1 kit (Molecular Probes) with the cationic fluorescent probe 5,5′,6,6′-tetrachloro-1,1′,3,3′-tetraethylbenzimidazolcarbocyanine iodide (JC-1), following the manufacturer’s instructions [19]. *T. vaginalis* trophozoites (5 × 10^5^ trophozoites/100 μL) were suspended in culture medium containing 100 μM CUR and incubated in 96-well cell culture plates for 2 h at 37 °C in a container under anoxic conditions. The probe was then added, and the plates were further incubated at 37 °C in darkness for 30 min. The plates were then centrifuged at 200× *g* for 5 min, to remove CUR, before being washed twice in assay buffer. Finally, fluorescence was measured in a microplate fluorimeter (Fx800, BioTek, Winooski, VT, USA) at excitation/emission wavelengths of 485/535 nm. All experiments were carried out in triplicate and a control without CUR was included to determine the fluorescence produced by spontaneous oxidation. This value was subtracted from the fluorescence emitted by the experimental samples.

### 4.5. Proteinase Purification 

The protocol was performed as described by [53], with slight modifications. The *T. vaginalis* trophozoites were washed three times with PBS and resuspended in equilibration buffer (100 mM CH_3_COONH_4_ and 10 mM CaCl_2_, pH 6.5). The samples were then sonicated on ice in a Branson W-250 sonifier (Branson Ultrasonic Corporation, Danbury, CT, USA), with 8 cycles of 10 pulses (duty cycle of 50% and out intensity 4), followed by centrifugation at 15,000× *g* for 10 min. The samples were then filtered (0.22 µm, Millipore, Burlington, MA, USA) and applied to a CNBr-activated bacitracin sepharose XK 16/20 column (GE Healthcare, Chicago, IL, USA) connected to a protein purification system (AKTAprim plus; GE Healthcare, USA). The column was washed with washing buffer (100 mM CH_3_COONH_4_, pH 6.5) until the 280 nm absorbance was basal. Elution buffer containing 100 mM CH_3_COONH_4_, 1 M NaCl and 25% (*v*/*v*) 2-isopropanol pH 6.5 was then added. Eluted samples were collected in 2.5 mL fractions until the OD at 280 nm was basal. Finally, samples were dialyzed against equilibration buffer, concentrated by ultrafiltration with Amicon Ultra 10 K centrifugal filter devices (Millipore, MA, USA) and stored in PBS 0.15 M at −80 °C. The protein concentration was determined by the method of Bradford [54] with a Bio-Rad Protein Assay kit (Bio-Rad Laboratories, Hercules, CA, USA) and with bovine serum albumin (Sigma–Aldrich) as standard [55].

### 4.6. SDS-PAGE-Gelatin Assay 

The effect of CUR on the activity of parasite proteinases was measured by sodium dodecyl sulphate polyacrylamide gel electrophoresis (SDS-PAGE) with 0.1% gelatin in 10% SDS linear gels under non-reduced conditions (without DTT) [53,55]. Parasites were cultured with or without a 50 µM CUR solution for 24 h and then washed twice in PBS and sonicated on ice in a Branson W-250 sonifier (Branson Ultrasonic Corporation, USA), with 8 cycles of 10 pulses (duty cycle of 50% and out intensity 4). The samples were then centrifuged at 20,000 × *g* for 15 min at 4 °C, and 6× loading buffer (containing 62 mM Tris–HCl buffer, pH 6.8, 2% SDS and 10% glycerol) was added to the resultant supernatant. The protein concentration in the extract was determined by the Bradford method [53], as described above. The samples were electrophoresed in a vertical electrophoresis system (Hoeffer, GE Healthcare, USA) for 45 min at 200 V with electrophoresis buffer (25 mM Tris, 190 mM glycine and 1% SDS, pH 8.3). The gel was cut into strips to separate different samples, which were then incubated for 30 min in a 2.5% Triton X-100 (*v/v*) solution and then in 0.1 M citrate buffer, pH 4 and 0.1 mM DTT, for 12 h at 37 °C with constant shaking, to enable development of the proteolytic activity of the *T. vaginalis* proteinases on the gelatin. Finally, the gels were stained with Thermo Scientific GelCode Blue Safe Protein Stain (Pierce, Thermo Fisher Scientific, Whalthan, MA, USA) to visualize the non-stained bands corresponding to lysed gelatin. Proteolytic bands appeared as clear bands on the stained blue background after destaining with water [53,55]. 

### 4.7. Gelatin-FITC Proteolytic Activity 

The quantitative effect of CUR on *T. vaginalis* proteolytic activity was assayed by measuring the hydrolytic capacity of the proteinases on a solution containing a protein substrate (gelatin) conjugated with a fluorescent ligand (fluorescein isothiocyanate; FITC). For the assay, FITC was dissolved with the gelatin in the same proportion in a Na_2_HPO_4_ solution at pH 9–9.5 and incubated for 1 h at room temperature. After gelatin and FITC coupling, excess FITC was removed by dialysis against PBS and centrifuged at 5000× *g* for 10 min to eliminate precipitated protein. Aliquots were stored at −20 °C until use. 

Aliquots (10 µL) of each lysate (sample with or without treatment with CUR), obtained as indicated above, were incubated with 20 µL of PBS and 20 µL of gelatin-FITC at 35 °C for 4 h in the dark in a moist chamber. After incubation, 120 µL of a 0.6 M trichloroacetic acid (TCA) solution was added and the mixture was held at room temperature for 30 min to enable precipitation of proteins. Finally, samples were centrifuged at 3000× *g* for 10 min and fluorescence was measured in a fluorimeter (Fl×800, Biotek, USA) at 488/520 nm excitation/emission. Three replicates of each sample were prepared.

### 4.8. Assay of Nitrite Production 

Production of NO by RAW 264.7 cells after culture for 48 h with *T. vaginalis* proteinases and CUR treatments was measured by the Griess reaction [56]. Supernatants from the cell cultures (100 μL) were incubated with 100 μL of Griess reagent containing 1% sulfanilamide, 0.1% *N*-(1-naphtyl)-ethylendiamine dihydrochloride and 2.5% H_3_PO_4_ at room temperature for 10 min. The absorbance was measured at 530 nm in a microplate reader (Titertek Multiscan, Flow Laboratories, Sunnyvale, CA, USA). The NO concentration was calculated with reference to a standard curve prepared with different concentrations of NaNO_2_ (1–200 μM in RAW 264.7 culture medium). Each treatment was prepared in triplicate. The NO synthase inhibitor LNMA (N(G)monomethyl-l-arginine monoacetate at 250 μM was added as a negative control of NO production and 100 ng/mL LPS was added as a positive control for NO production [52,57].

### 4.9. Real-Time Quantitative Reverse Transcriptase-Polymerase Chain Reaction (RT-qPCR) 

The RT-qPCR assay was performed as previously described [19], with minor modifications. After treatment for 24 h with CUR and/or lipopolysaccharide (LPS) and proteinases (in the case of macrophages), total RNA from *T. vaginalis* trophozoites (10^7^ cells/sample) or from RAW 264.7 cells were isolated with NucleoSpin RNA kit (Macherey-Nagel, Düren, Germany), following the manufacturer’s instructions. Purified RNA was treated with DNase I (RNase free, Thermo Fisher Scientific, Whalthan, MA, USA) and the purity and final concentration were estimated in a NanoDrop ND-1000 spectrophotometer (Thermo Fisher Scientific). cDNA was synthesized (25 µL/reaction mixture) with 1.25 μM random hexamer primers (Roche, Basel, Switzerland), 250 μM deoxynucleosides triphosphate (dNTPs), each, 10 mM dithiothreitol (DTT), 20 U of RNase inhibitor, 2.5 mM MgCl_2_ and 200U of MMLV (Moloney murine leukemia virus) reverse transcriptase (Promega, Madison, WI, USA) in reaction buffer containing 30 mM Tris and 20 mM KCl (pH 8.3). Two micrograms of RNA was added per sample.

The qPCR reaction for *T. vaginalis* samples was performed with gene-specific primers for the *pyruvate-ferredoxin oxidoreductase D* (*pfoD*) gene (forward/reverse primer pair 5‘-TCTCCGTTCTTGATCGTTCC-3′/5′ -TGTTGTCGAAGACAGCCTG-3′) and for the *β-tubulin* (*Tub*) gene (forward/reverse primer pair 5′-TACTCCATCGTCCCATCTCC-3′/5′ -CCGGACATAACCATGGAAAC-3′), used as a housekeeping gene, to normalize data.

For mice expression assays, the following primer sequences were used: tumor necrosis factor α (TNFα): forward/reverse primer pair 5′-AGCCCCCAGTCTGTATCCTT -3′/5′- CTCCCTTTGCAGAACTCAGG-3′, interleukine-1β (IL-1β) forward/reverse primer pair 5′- GCCCATCCTCTGTGACTCAT-3′/5′-AGGCCACAGGTATTTTGTCG -3′; glucocorticoid receptor (mGC) forward/reverse primer pair 5′- AGGCCGCTCAGTGTTTTCTAA-3′/TTACAGCTTCCACACGTCAGC-3′; chaperones Hsp70 (Hsp70) forward/reverse primer pair 5′- CATCATCAATGAGCCCACAG-3′/5′-TCTTGTGTTTGCGCTTGAAC-3′ and β-actin (ACT) forward/reverse primer pair 5′-AGCCATGTACGTAGCCATCC-3′/5′-CTCTCAGCTGTGGTGGTGAA-3′; ACT was used to normalize data, which were expressed in relative arbitrary units (RLU). 

Primer pairs were designed and optimized with the Primer 3 Plus program (http://www.bioinformatics.nl/cgibin/primer3plus/primer3plus.cgi) with a Tm of 60 °C The values show the mean ± the standard error (SE) of the relative expression in arbitrary units of three trials.

The qPCR mixtures (10 μL) were prepared with a reaction mix already containing the assay buffer, dNTPs and SYBR Green (Maxima SYBR Green qPCR Master Mix, Thermo Scientific). Primer pairs were used at a final concentration of 300 nM and 1 µL of cDNA was added per well, and the 10 μL final volume was completed with RNAse free distilled H_2_O.

The qPCR reaction was conducted at 95 °C for 5 min followed by 40 cycles of 10 s at 95 °C and 30 s at 60 °C. Melting curve analysis was then carried out at 95 °C for 15 s, 55 °C for 15 s, and 95 °C for 15 s. The qPCR product specificity and size were confirmed by gel electrophoresis with 2% agarose [19]. All reactions were performed in a real time PCR system, Eco Real-time PCR system (Illumina, San Diego, CA, USA). Relative quantification of gene expression was determined by 2^−ΔΔCq^ method [58] in accordance with MIQUE (minimum information guidelines for publishing real-time quantitative PCR experiments) [59].

### 4.10. Chemiluminiscent Enzyme-Linked ImmunoSorbent Assay (ELISA) 

An indirect ELISA was used to detect mGR and Hsp70 antibodies against the crude extract (CE) of RAW 267.4 cells as previously described [60], with minor modifications. The cells were incubated with the different treatments (CUR at 100 μM, lipopolysaccharide; LPS at 100 ng/mL and proteinases 50 μg/mL) for 24 h, collected by scraping, centrifuged at 250× *g* for 5 min and rinsed in PBS. Ten volumes of Cell Lysis (ProteoJET™ Cytoplasmic and Nuclear Protein Extraction Kit) containing protease inhibitor cocktail (ProteoBlock™ Protease Inhibitor Cocktail) were added followed by 0.1 M DTT. The mixture was vortexed for 10 sec, placed on ice for 10 min and vortexed again. Samples were diluted in carbonate–bicarbonate buffer (pH 9.6) and the protein concentration was estimated by the Bradford method, as described above. The antigen (1.5 μg) was then added to 96-well ELISA plates (Iwaki Europe GmbH, Willich, Germany) and incubated overnight at 4 °C. The plates were washed three times with TBS (50 mM Tris, 0.15 M NaCl, pH 7.4) and blocked for 1 h with TBS containing 0.2% Tween 20 (TBS-T1) and 5% non-fat dry milk. The plates were then incubated in a microplate shaker at 750 rpm for 30 min at room temperature with the antibodies (rabbit anti-GR (M-20): sc-1004 and rabbit anti-HSP 70/HSC 70, H-300, Santa Cruz Biotechnology, New York, NY, USA) in a 1:500 dilution (in TBS-T1 containing 1% non-fat dry milk) and washed five times with TBS containing 0.05% Tween 20. Bound antibodies were detected with peroxidase (HRP)-conjugated goat anti-rabbit Ig (Dako, Agilent, Santa Clara, CA, USA) diluted 1:1000 in TBS-T1 and incubated in a microplate shaker at 750 rpm for 30 min. The plates were washed five times in TBS before addition of 100 μL to each well of enhanced luminol-based chemiluminiscent substrate for detection of HRP (Thermo Fisher Scientific) After 2 min of incubation, luminescence was measured in a multi-detection microplate reader (Biotek, FL×800) in the configuration for luminescence measurements.

### 4.11. Statistical Analysis

Results are expressed as mean values ± standard error of means (SEM). Data were tested by one-way analysis of variance (ANOVA) followed by a Tukey–Kramer test for multiple comparisons, and differences were considered significant at *p* ≤ 0.05.

## Figures and Tables

**Figure 1 molecules-25-05321-f001:**
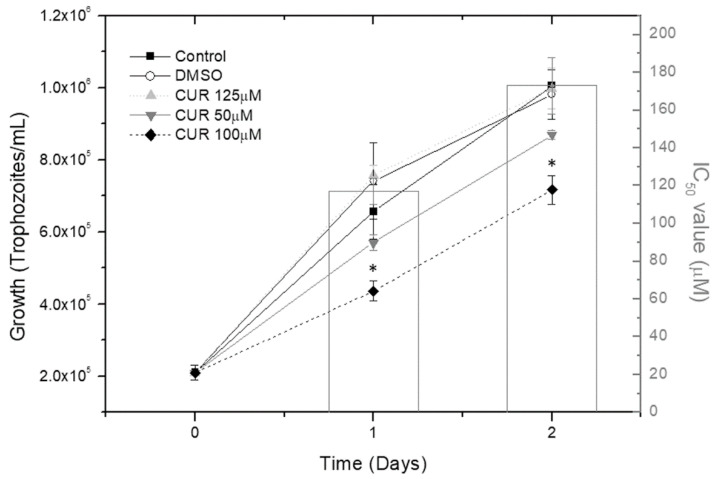
Antiparasitic effect of curcumin (CUR). Growth rate of *T. vaginalis* trophozoites cultured in vitro in MDM with different doses of CUR (50, 100 and 125 μM) at 35 °C in a container under vacuum. The highest concentration of DMSO used in CUR solutions, was added alone to some wells, as a control. Data were obtained by counting the cells with a hemocytometer on 2 consecutive days (*n* = 5) (* *p* < 0.05). Bars indicate the IC_50_ ± standard error on day 1 (117 ± 7 μM) and on day 2 (173 ± 15 μM) of culture.

**Figure 2 molecules-25-05321-f002:**
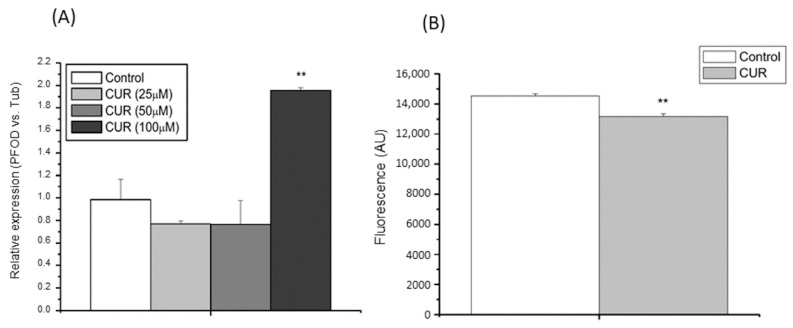
Effect of curcumin (CUR) on energy metabolism in *T. vaginalis*. (**A**) Effect of CUR on the *T. vaginalis* hydrogenosomal enzyme pyruvate-ferredoxin oxidorreductase (PfoD) mRNA expression levels, as revealed by RT-qPCR. Data are expressed relative to *T. vaginalis* β-tubulin mRNA expression levels (*n* = 3) (** *p* < 0.01). (**B**) Effect of CUR on hydrogenosomal membrane potential (ΔΨm), as revealed by JC-1 fluorescent probe, on *T. vaginalis* trophozoites incubated for 2 h under microaerobic conditions in MDM with 100 μM CUR. Untreated *T. vaginalis* trophozoites were included as control. Data are expressed in fluorescence arbitrary units (AU). (*n* = 5) (** *p* < 0.01).

**Figure 3 molecules-25-05321-f003:**
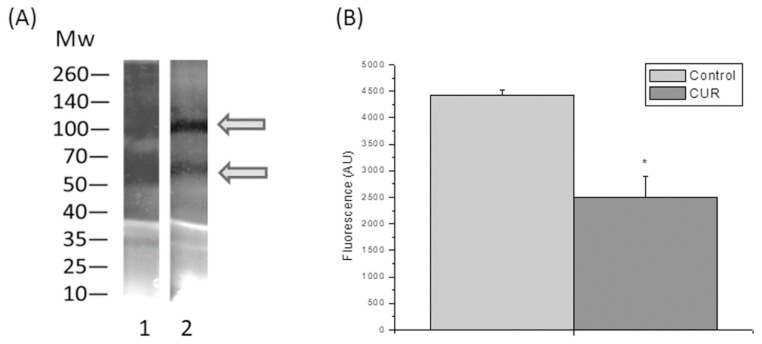
Effects of curcumin (CUR) on parasite proteolytic activity. (**A**) SDS–PAGE-gelatin analysis of protease activity in *T. vaginalis* total protein extracts treated for 24 h with (2) or without (1) CUR (50 µM). Gel strips were incubated for 12 h at 37 °C with DTT. Proteolytic activity is shown in black in the picture. Arrows indicate bands showing a decrease in proteolytic activity relative to untreated samples. Mw, molecular weight markers (kDa). (**B**) Quantitative analysis of proteolytic activity measured by a fluorimetric assay with gelatin-FITC as an enzymatic substrate. Data are expressed in fluorescence arbitrary units (AU) (*n* = 5, * *p* < 0.05).

**Figure 4 molecules-25-05321-f004:**
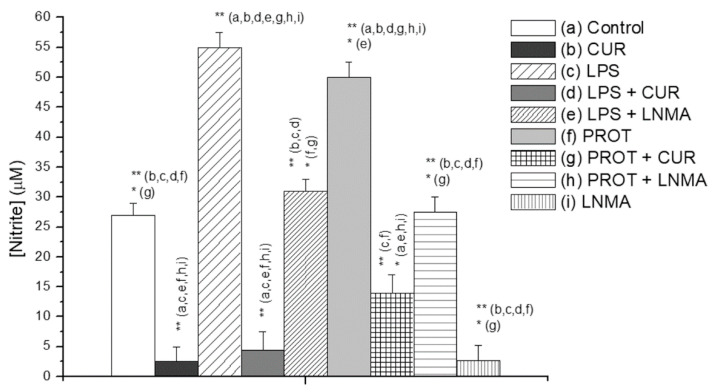
Effect of curcumin (CUR, 100 μM) on nitric oxide (NO) production. NO production was determined in RAW 267.4 cells after 24 h incubation with the proinflammatory stimuli lipopolysaccharide (LPS, 100 ng/mL) and proteases from *T. vaginalis* (50 μg/mL). Data represent the nitrite concentration in µM units, measured by spectrophotometry and extrapolated from a regression line constructed for known concentrations of nitrite. The inhibitor of NO synthase LNMA was added (at 250 μM) to control for the inhibitory effect; (*n* = 5) (* *p* < 0.05; ** *p <* 0.01).

**Figure 5 molecules-25-05321-f005:**
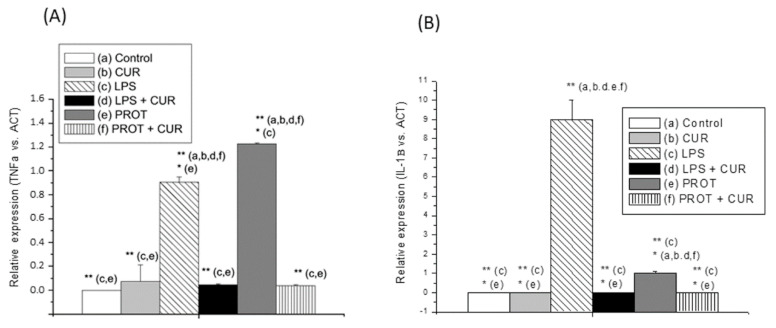
Relative mRNA expression of the genes of proinflammatory cytokines, quantified by RT-qPCR. Relative expression of RAW 267.4 cells TNFα (**A**) and IL-1β (**B**) genes after 24 h incubation with *T. vaginalis* proteases, and LPS or CUR. Data are expressed in relative expression units normalized against the housekeeping α-actine (ACT) gene mRNA levels (*n* = 3). Asterisks indicate statistically significant differences (* *p* < 0.05; ** *p* < 0.01) relative to the control. CUR: curcumin 100 µM; LPS: lipopolysaccharide 100 ng/mL; PROT: proteases 100 µg/mL.

**Figure 6 molecules-25-05321-f006:**
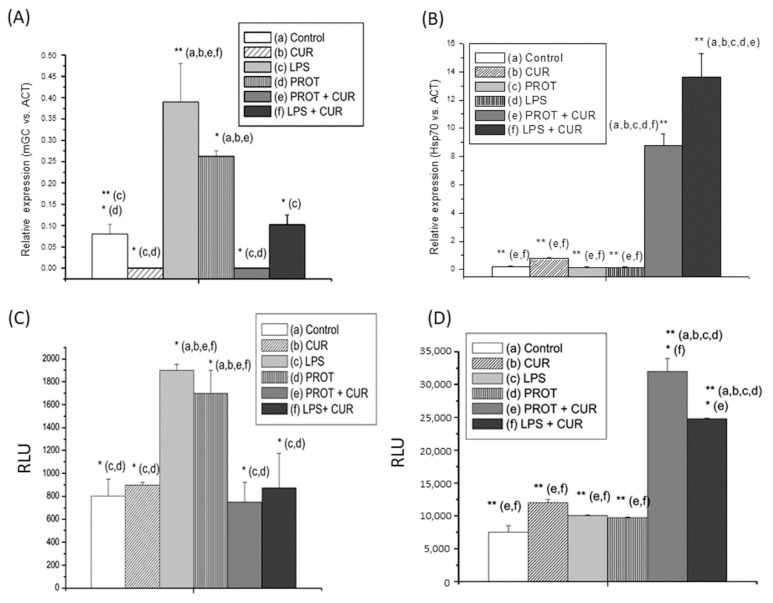
Influence of curcumin (CUR) on expression of glucocorticoid receptor (mGR) and the chaperone heat shock protein 70 (Hsp70). Assays were performed by RT-qPCR (RNA) or ELISA (RAW 267.4 cell protein extracts) and obtained after incubation for 24 h with the different treatments: *T. vaginalis* proteinases (PROT (50 µg/mL)), lipopolysaccharide (LPS (100 ng/mL)) and CUR (100 µM). Data indicate inhibition of the upregulation of mGR protein and mRNA expression levels driven by parasite proteinases in the presence of CUR (**A**,**C**, respectively). Nevertheless, expression of Hsp70 protein and mRNA increased in the presence of the polyphenol (**B**,**D**, respectively). For RT-qPCR assays, data are expressed in relative values normalized relative to actin (ACT) gene expression (*n* = 3) (* *p* < 0.05; ** *p* < 0.01). For ELISA assays, all data are expressed as means ± standard errors of luminescence measured in relative light units (RLU) (*n* = 3). * *p* < 0.05; ** *p* < 0.01.

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
