# Peer review of "Molecular Targets Implicated in the Antiparasitic and Anti-Inflammatory Activity of the Phytochemical Curcumin in Trichomoniasis"

_molecules, 2020, doi:10.3390/molecules25225321_

Round 1
Reviewer 1 Report
Dear Authors,
In fact, we observe continuous progress in the discovery and development of new drugs by using biologically active compounds obtained from medicinal plants, including curcumin. This creates new perspectives for improving pharmacotherapy, and even designating new therapeutic solutions based on the assessed mechanisms of curcumin's action. The authors' study is very important for pharmacotherapy in trichomoniasis and brings new light to molecular targets implicated in the antiparasitic and anti-inflammatory activity of curcumin. The authors searched for molecular targets on which CUR can act and that explain the antiparasitic effect of the compound. They also investigated the influence of CUR on molecules that are key to parasite survival (enzymes implicated in hydrogenosomal metabolism) and host invasion (proteinases). In addition, they investigated the modulatory role of CUR on the pro-inflammatory response induced by T. vaginalis proteinases in a mice macrophage model. The presented studies were comprehensive and complemented each other.
These following concerns need to be addressed:
1) At line 180 – The authors should correct concentration unit of CUR (in brackets)
2) At lines 285-286 – The authors should correct information in sentence
i.e. “The in vitro half maximal inhibitory concentrations (IC50) of CUR (in relation to the antiparasitic activity) and the relative gene expression after treatment of CUR were calculated by regression analysis”.
What genes did the Authors mean in this point?
I recommend this paper for publication after minor revisions.
Best regards
Author Response
Reviewer nº 1:
These following concerns need to be addressed:
Point 1) At line 180 – The authors should correct concentration unit of CUR (in brackets)
Response: This error has been corrected in the final version.
Point 2) At lines 285-286 – The authors should correct information in sentence
i.e. “The in vitro half maximal inhibitory concentrations (IC50) of CUR (in relation to the antiparasitic activity) and the relative gene expression after treatment of CUR were calculated by regression analysis”.
What genes did the Authors mean in this point?
Response: Indeed, the IC50 calculation was carried out by means of a regression analysis only for the determination of its antiparasitic activity, and not for the expression of the genes, which was carried out as described in lines 391-393 of the original manuscript.
Reviewer 2 Report
The authors report their finding in the Molecular targets implicated in the antiparasitic and anti-inflammatory activity of curcumin in trichomoniasis.
Some comments:
Among quite a lot known compounds in Curcuma longa, why the authors only choose curcumin in their study? The authors may explain in the introduction.
At 100uM of curcumin in their RAW cells study, this is a relative high concentrations and it may inhibit the growth of cells. I suggest the authors should do a cytotoxicity tests of curcumin on RAW cells.
Author Response
Reviewer nº 2:
Comments and Suggestions for Authors
The authors report their finding in the Molecular targets implicated in the antiparasitic and anti-inflammatory activity of curcumin in trichomoniasis.
Some comments:
Point 1) Among quite a lot known compounds in Curcuma longa, why the authors only choose curcumin in their study? The authors may explain in the introduction.
Response: We have included in the introduction a paragraph that provides additional information on the composition of turmeric and an explanation of why curcumin was used specifically in this study:
The rhizomes of Curcuma longa include curcuminoids (polyphenolic pigments) such as curcumin, demethoxycurcumin and bis-methoxycurcumin, as well as volatile oils, sugars, proteins and resins; however, curcumin (diferuloylmethane) is generally the main curcuminoid found in turmeric, and generally considered its most active constituent to which the main pharmacological activities are attributed (Jurenka, 2009).
Point 2) At 100 μM of curcumin in their RAW cells study, this is a relative high concentrations and it may inhibit the growth of cells. I suggest the authors should do a cytotoxicity tests of curcumin on RAW cells.
Response: In accordance with the referee's instructions, we have performed a curcumin viability / cytotoxicity assay using an MTT assay. The results obtained have been included as supplementary material and, as can be seen in the figure included in that section, although the curcumin concentration of 100 µM produces a slight decrease in cell viability; however, it is not statistically significant. In contrast, when concentrations exceed 100 µM and approach concentrations that are cytotoxic to T. vaginalis (117-173 µM), curcumin is also cytotoxic to RAW 264.7 cells.
Round 2
Reviewer 2 Report
No further comment.